# Neutrophil Extracellular Traps Promote Platelet-Driven Contraction of Inflammatory Blood Clots via Local Generation of Endogenous Thrombin and Softening of the Fibrin Network

**DOI:** 10.3390/cells14242018

**Published:** 2025-12-18

**Authors:** Shakhnoza M. Saliakhutdinova, Rafael R. Khismatullin, Alina I. Khabirova, Rustem I. Litvinov, John W. Weisel

**Affiliations:** 1Department of Morphology and General Pathology, Institute of Fundamental Medicine and Biology, Kazan Federal University, 420015 Kazan, Russia; abdullayevashahnoza026@gmail.com (S.M.S.); rafael.khismatullin@gmail.com (R.R.K.); alina.urussu.95@gmail.com (A.I.K.); 2Department of Cell and Developmental Biology, University of Pennsylvania School of Medicine, Philadelphia, PA 19104, USA; litvinov@pennmedicine.upenn.edu

**Keywords:** neutrophils, neutrophil extracellular traps, clot retraction, immunothrombosis

## Abstract

**Highlights:**

**What are the main findings?**
Activated neutrophils promote blood clot contraction, and this effect is associated with formation of neutrophil extracellular traps (NETs) embedded in the fibrin network.NETs stimulated clot contraction by enhancing the production of endogenous thrombin and reducing the stiffness of blood clots.

**What is the implication of the main finding?**
The results obtained provide novel mechanistic insights into the pathophysiology of inflammatory thrombosis, suggesting that the presence of NETs creates a unique thrombus phenotype with enhanced contractility.This finding may have important implications for understanding thrombotic complications in sepsis, COVID-19, and other inflammatory conditions, potentially guiding the development of future therapeutic strategies.

**Abstract:**

Immunothrombosis can substantially affect the course and outcomes of severe infections and immune-mediated diseases. While inflammatory thrombi are neutrophil-rich, impact of neutrophils on clot contraction, a key modulator of thrombus stability and obstructiveness, was unknown. This study investigated how neutrophils and neutrophil extracellular traps (NETs) affect the rate and extent of platelet-driven clot contraction. Isolated human neutrophils were stimulated with phorbol-12-myristate-13-acetate (PMA) to induce NETosis, confirmed by fluorescence microscopy and scanning electron microscopy. Thrombin-induced clots, formed from whole blood or platelet-rich plasma, were supplemented with non-activated or PMA-activated neutrophils. Clot contraction kinetics and viscoelasticity were analyzed. PMA-activated neutrophils significantly enhanced the rate and final extent of clot contraction compared to controls. This promoting effect was abolished by deoxyribonuclease (DNAse) I, confirming that it was mediated by NETs embedded in the fibrin network. The factor Xa inhibitor rivaroxaban also abrogated this effect, indicating a role for NET-induced endogenous thrombin generation and platelet hyperactivation. Thromboelastography revealed that NETs made clots softer and more deformable. We conclude that activated neutrophils promote clot contraction via NETs embedded in the fibrin network, which enhance platelet contractility via endogenous thrombin production and increase clot deformability, suggesting that inflammatory thrombosis may require treatments addressing this enhanced contractility.

## 1. Introduction

Inflammatory thrombosis, or immunothrombosis, represents a pathophysiological intersection between inflammation and blood clotting, playing an important role in various infectious and immune-mediated inflammatory diseases with a relatively high prevalence globally and remaining a significant cause of illness and death [1,2,3]. Immunothrombosis is associated with the activation of immune cells, particularly neutrophils and monocytes, which contribute to thrombus formation through the release of procoagulant agents and formation of neutrophil extracellular traps (NETs) [4,5,6].

An important yet understudied aspect of thrombosis is the process of platelet-driven clot contraction (retraction), which has long been recognized in vitro as a manifestation and measure of platelet functionality [7,8,9], but it also occurs in vivo as a part of hemostasis and thrombosis [10,11]. Platelet-driven contraction modulates key intravascular thrombus properties, including obstructiveness, susceptibility to lysis, stiffness, porosity, and resistance to rupture, all of which determine clinical course and outcomes of thrombosis [12]. However, how blood clot contraction is linked to immunothrombosis remains unclear.

It is known that clot contraction is highly dependent on thrombus cellular composition, in particular the number of platelets and volume fraction of red blood cells (RBCs) [13]. Given that inflammatory thrombi are enriched with neutrophils, their presence likely alters contraction dynamics compared to conventional non-inflammatory thrombi. Activation of neutrophils is accompanied by NETosis, the extrusion of nuclear chromatin components, in particular deoxyribonucleic acid (DNA) and histones (NETs), under the influence of inflammatory stimuli [4,14]. NETs formation is known to enhance blood clotting and promote platelet activation, but the real effects of NETs on blood clot contraction remain unknown.

This study fills a gap in understanding the interplay between immunothrombosis and blood clot contraction. The specific aim of the current work was to study the effects of the activated neutrophils on the rate and degree of platelet-mediated contraction of blood clots. We have shown that activated neutrophils promote blood clot contraction, and this effect is associated with formation of extracellular traps embedded in the fibrin network. NETs stimulated clot contraction by enhancing the production of endogenous thrombin, as well as affecting the mechanical properties of blood clots.

## 2. Materials and Methods

### 2.1. Blood Collection and Fractionation

All procedures involving human subjects were approved by the Ethical Committee of Kazan Federal University (28 December 2020, protocol #27; renewed 26 February 2025, protocol #53). Written informed consent was obtained from healthy donors enrolled in the study. For in vitro model experiments, blood was drawn from 30 donors, 20 (67%) men and 10 (33%) women, aged 18 to 36 years (average 26 ± 4 years), excluding those who smoke, have recently had infectious diseases or had taken anti-inflammatory drugs or anticoagulants within 2 weeks prior to blood drawing. Blood collection and handling adhered to approved guidelines and followed standard pre-analytical requirements. Blood was collected by venipuncture into 3.2% trisodium citrate and mixed 9:1 by volume in vacutainers (Vacuette, Greiner Bio-one, Monroe, NC, USA). One portion of citrated blood samples was used for neutrophil isolation (see below). The second portion was centrifuged (200× *g*, 10 min, room temperature) to obtain platelet-rich plasma (PRP). A portion of PRP was centrifuged at 2000× *g* for 10 min to obtain platelet-poor plasma (PPP) and a part of PPP was centrifuged at 10,000× *g* for 5 min to obtain platelet-free plasma (PFP). Whole blood and its fractions were used no later than 4 h after blood collection.

### 2.2. Isolation and Activation of Neutrophils to Produce Neutrophil Extracellular Traps (NETs)

Neutrophils were isolated from fresh citrated blood (5 mL) layered on 5 mL of the Lympholyte-Poly Cell Separation Media (Cedarlane, Burlington, ON, Canada) by centrifugation at 500× *g* for 35 min at room temperature. Following centrifugation, the opaque layer containing neutrophils was collected, and the cells were washed three times with Ca^2+^- and Mg^2+^-free Hank’s balanced salt solution (HBSS) by resuspension and centrifugation for 10 min at 300× *g*. After the last wash, the cells were resuspended in 250 μL of HBSS. The number of cells (including platelet count), purity of neutrophil preparations, and cell morphology were assessed microscopically in a hemocytometer, flow cytometry, and scanning electron microscopy, respectively (see below). A portion of the isolated neutrophils was activated by adding 100 nM (final concentration) phorbol-12-myristate-13-acetate (PMA) and incubating 3.5 h at 37 °C in a CO_2_ incubator (New Brunswick Galaxy 170R, Marshall Scientific, Hampton, NH, USA) to induce the release of NETs. In further experiments, two sources of NETs were used: (i) suspension of PMA-activated neutrophils and (ii) supernatant of PMA-activated neutrophils formed after gravitational sedimentation of the cells. Typically, 55 μL of the PMA-activated neutrophil suspension or supernatant containing 100 nM PMA was added to 200 μL of whole blood or PRP samples, so that the residual final concentration of PMA was about 22 nM.

### 2.3. Flow Cytometry of Isolated Neutrophils

To assess purity and functional state of neutrophils, flow cytometry with fluorescently labeled monoclonal antibodies was used. Isolated neutrophils, either non-activated or PMA-activated, were incubated for 10 min in the dark in the presence of two types of labeled antibodies: (i) anti-human CD16 (BD Biosciences, San Jose, CA, USA) used as a neutrophil-specific marker; (ii) anti-human-activated CD11b (Biolegend, San Diego, CA, USA) used as a marker of neutrophil activation followed by NETosis since CD11b is the αM subunit of the αMβ2 (Mac-1) integrin. Anti-CD16 antibodies were labeled with phycoerythrin (PE), while anti-CD11b antibodies were labeled with allophycocyanin (APC). Flow cytometry was conducted using either a CytoFlex (Beckman Coulter, Brea, CA, USA) or FACSCalibur (Beckman Dickinson, Franklin Lakes, NJ, USA) instruments, and data were analyzed using FlowJo software vX.0.7 (Beckman Dickinson, USA). For each measurement, 5000 neutrophils were analyzed. Neutrophils were gated using forward (FSC) and side scatter (SSC) and quantified further as CD16+ signals. Within a CD16+ gate, activated neutrophils were identified as CD11b+ events and presented as a fraction of CD16+ signals taken as 100%.

### 2.4. (Immuno)Histochemical Examination of PMA-Activated Neutrophils and NETs

For histological studies, PRP clots containing non-activated or PMA-activated neutrophils were formed in a tube pre-lubricated with 1% Pluronic F-127 (MilliporeSigma, Burlington, MA, USA) to prevent the clot sticking to the walls. A PRP sample containing ~48 × 10^6^ platelets was added to ~0.8 × 10^6^ neutrophils in HBSS (~60:1 physiological cell type ratio) to a final volume of 500 μL, and clot formation was initiated by the addition of 2 mM CaCl_2_ and 1 U/mL thrombin (final concentrations). After incubating for 30 min, the clots were fixed in 10% neutral buffered formalin for 1 h, then rinsed with water, treated with isopropanol and xylene in a tissue processor (STP420ES, Thermo Scientific, Waltham, MA, USA), and embedded in paraffin. Then, 4 µm thick sections were deparaffinized and stained with a hematoxylin and eosin histochemical kit (Thermo Scientific, USA). Another set of 4 μm thick slices was used for immunohistochemical staining. For live confocal microscopy of hydrated unfixed samples, PRP clots containing PMA-activated neutrophils were formed in confocal dishes and subjected to immunofluorescent analysis. The same staining techniques were used to visualize cells and NETs in clots and isolated PMA-activated neutrophils.

For the peroxidase method of antigen visualization, clot sections were first deparaffinized and rehydrated using standard techniques. For antigen retrieval, the slices were boiled for 30 min in 0.01 M citrate buffer (pH 6.0), left in the same buffer for another 15–20 min, and then transferred to Dulbecco’s phosphate-buffered saline (DPBS). To block endogenous peroxidase activity, the samples were incubated for 20 min in a 3% hydrogen peroxide solution in DPBS. To reduce non-specific binding, all preparations were incubated for 2 h in 200 μL of a blocking solution containing 10% donkey serum (d9663, Sigma-Aldrich, St. Louis, MO, USA) and 1% bovine serum albumin (Dia-M, Southfield, MI, USA) in DPBS. The next step was overnight incubation with primary rabbit antibodies against human citrullinated histones H3 (R2 + 8 + 17, H3Cit; cat. # ab5103; Abcam, Cambridge, UK) diluted 1:400 in DPBS containing 1% BSA. For control, clots were formed in the absence of neutrophils, without primary antibodies (DPBS instead of antibodies) or by adding mouse IgG1 for isotype control. After washing with DPBS, the sections were incubated with secondary biotinylated anti-rabbit antibodies (VECSTATIN ABC-HRP Kit, Vector Laboratories, Newark, CA, USA) for 30 min, washed and incubated with the peroxidase-containing solution of the kit for 30 min. The washed slices were processed using 3,3′-diaminobenzidine (DAB) solution (K3467, Dako, Carpinteria, CA, USA), a chromogenic peroxidase substrate. Alternatively, fluorescently labeled secondary antibodies (donkey anti-rabbit IgG (H + L), Alexa Fluor™ 488, A21206, Invitrogen, Carlsbad, CA, USA) were used at a 1:400 dilution and incubated for 1 h in the dark. To visualize DNA, the clot was washed in DPBS, fixed for one hour in 10% buffered paraformaldehyde, and then subjected to DNA-specific 4′,6-diamidino-2-phenylindole (DAPI, BioLegend, USA) staining in the dark. Histological imaging was performed using a Hamamatsu NanoZoomer 560 scanner (Hamamatsu Photonics, Hamamatsu, Japan); confocal fluorescent light microscopy was performed using a laser scanning confocal microscope LSM 780 (Carl Zeiss, Oberkochen, Germany).

### 2.5. Scanning Electron Microscopy of Neutrophils and NETs

For high-resolution scanning electron microscopy, the following samples were prepared: (i) isolated non-activated and PMA-activated neutrophils; (ii) supernatant of PMA-activated neutrophils; (iii) thrombin-induced PFP clots formed in the presence of PMA-activated neutrophils. It is noteworthy that PRP was not used for scanning electron microscopy because the presence of platelets interferes with the visualization of NETs and their interaction with fibrin fibers. Fixed isolated neutrophils and the supernatant were layered on coverslips pre-treated with 0.01% poly-L-lysine (EMD Millipore, Burlington, MA, USA) overnight and incubated for 30 min for attachment of cells and NETs on a cover glass followed by fixation in 2% glutaraldehyde for 90 min at room temperature. Thrombin-induced PFP clots in the presence of activated neutrophils (see above) were allowed to form on coverslips for 20 min and then fixed in 2% glutaraldehyde for 90 min at room temperature. All the fixed samples were rinsed three times with a 50 mM cacodylate buffer/150 mM NaCl, pH 7.4, for 10 min, dehydrated in graded ethanol, immersed in hexamethyldisilazane, and air-dried. A layer of gold palladium made using a sputter coater Quorum Q 150T ES (Quorum, Lewes, UK). Images were taken with a scanning electron microscope (Merlin, Carl Zeiss, Germany).

### 2.6. Blood Clot Contraction Assay

Contraction of thrombin-induced clots in whole citrated human blood or PRP was tracked optically as a reduction in clot size in the absence or presence of non-activated or PMA-activated neutrophils. In the control samples, the corresponding volume of HBSS buffer was added. The kinetics of clot contraction were followed using the Thrombodynamics Analyzer System (HemaCore Ltd., Moscow, Russia). To study effects of isolated non-activated or PMA-activated neutrophils, 50–80 μL HBSS containing ~8 × 10^5^ neutrophils was added to 120–150 μL of blood or PRP to the final volume of 200 μL. The samples were activated with 2 mM CaCl_2_ and 1 U/mL human thrombin (Sigma-Aldrich, USA) (final concentrations). Before clot formation, blood samples (80 µL) containing thrombin were quickly placed into transparent plastic cuvettes (12 mm × 7 mm × 1 mm), pre-lubricated with 4% *v*/*v* Triton X-100 in 150 mM NaCl for blood or 1% Pluronic for PRP to prevent clot adhesion to the walls of the chamber, and the cuvette was placed at 37 °C in the optical analyzer. Digital images of the contracting clots were taken on scattered light every 15 s for 20 min. The changes in clot size were plotted computationally as a kinetic curve and the following parameters of clot contraction were extracted: (i) the final extent of contraction calculated as a difference between the initial and end clot size at the 20 min point and expressed as percentage of the initial size; (ii) lag time (in seconds) determined as the time until the extent of contraction reaches 5%; (iii) the average contraction rate, which is a final extent of contraction divided by the time; and (iv) the area under the kinetic curve of contraction (a.u.), an integral characteristic of the intensity of clot contraction.

### 2.7. Thromboelastography (TEG)

Thromboelastograms were recorded using the TEG 5000 instrument (Haemoscope, Niles, IL, USA). Citrated blood or PRP samples were mixed with either a physiological buffer (control) or isolated neutrophils, non-activated or PMA-activated. Next, CaCl_2_ (final concentration 2 mM) and human thrombin (Sigma-Aldrich, USA) (final concentration 1 U/mL) were added, after which 360 μL of the mixture was quickly transferred into a TEG cuvette preheated to 37 °C, and the measurement was started. After recording a thromboelastogram, the following TEG parameters were determined: *R*—reaction time, i.e., the time from the initiation of clotting to the onset of fibrin formation; *G’*—the storage (elastic) modulus of a clot calculated from the maximal amplitude (*MA* in mm) according to the manufacturer’s formula: *G’* (dyn/cm^2^) = [5000 × *MA*/(100 − *MA*)].

### 2.8. Statistical Analysis

GraphPad Prism 8.0.1 (GraphPad Software, San Diego, CA, USA) software package was used. Parametric or non-parametric distributions were assessed using the Shapiro–Wilk and D’Agostino–Pearson criteria. Statistical significance for paired data was estimated with the parametric Student’s *t*-test or nonparametric Mann–Whitney U test or Wilcoxon tests. Multiple comparisons were analyzed using one-way ANOVA with the post hoc Sidak’s test. Multi-group column Repeated-Measures analysis for data with non-normal distribution was performed using the nonparametric Friedman test. The level of significance was 95% (*p* < 0.05).

## 3. Results

### 3.1. Activated Neutrophils Promote Clot Contraction

To mimic major aspects of thromboinflammation, we studied effects of PMA-activated neutrophils on clot contraction in comparison with non-activated neutrophils added to whole blood or PRP at the amount corresponding to a physiological neutrophil count (8000/μL). The averaged kinetic curves of clot contraction in whole blood (Figure 1A) and PRP (Figure 1B) in the absence (control) and presence of non-activated or PMA-activated neutrophils showed that the final extent of clot contraction increased in the presence of PMA-activated neutrophils. Numerical data analysis confirmed that there was a significant difference in the final extent of clot contraction in the presence of PMA-activated neutrophils versus non-activated neutrophils and control samples without addition of exogenous neutrophils in both whole blood (Figure 1C) and PRP (Figure 1D).

Although the kinetics of clot shrinkage in the curves look somewhat similar, the overall dynamics reflecting the rate of contraction were significantly greater in the presence of the PMA-activated neutrophils (*p* < 0.0001, Friedman test). The numerical values of the parameters studied and statistical analysis are presented in Appendix A. Notably, there was a significant decrease in the average lag time of contraction in whole blood clots and an increase in the average rate of contraction in PRP in the presence of PMA-activated neutrophils compared to non-activated neutrophils and neutrophil-free control samples (Figure 2, Appendix A). Other kinetic parameters of whole blood clot contraction also showed a trend toward acceleration of clot contraction by the PMA-treated neutrophils, but they were statistically insignificant. Taken together, these differences suggest accelerated platelet activation [15] and increased contractility in the presence of activated neutrophils.

To exclude the possibility that the observed stimulating effects of PMA-treated neutrophils on clot contraction were due to direct effects of the residual PMA on platelets or other blood cells, we performed a control experiment in which PMA alone (without neutrophils) was added to PRP in the same final (residual) concentration (~22 nM) as with the PMA-treated neutrophil preparations (the cell-activating PMA concentration was 100 nM). The results showed that PMA, at this relatively low concentration and within a limited time frame, affected neither the rate nor the final extent of clot contraction, remaining at the level of 80–83% (Figure 3, Appendix A). Therefore, a potential artifact associated with the contamination of neutrophils with PMA has been excluded.

### 3.2. Activated Neutrophils Affect the Phase Kinetics of Clot Contraction

It is known that contraction of blood clots has three kinetic phases: initiation of contraction, linear contraction, and mechanical stabilization, designated as phases 1, 2, and 3, respectively [13]. Here, we performed the same type of phase analysis for individual kinetic curves of clot contraction in whole blood and PRP averaged in Figure 4.

In both whole blood and PRP clots, in the presence of PMA-activated neutrophils, the duration of the initiation phase (phase 1) increased (Figure 4; Appendix A). In combination with the greater final extent of clot contraction in the presence of PMA-activated neutrophils compared to non-activated neutrophils (Figure 1), this finding suggests a key and dominant role of phase 1 reactions in the overall enhancement of clot contraction caused by the PMA-activated neutrophils. Considering that phase 1 reflects platelet activation caused by exogenous and endogenous thrombin, the observed extension of the first phase in the presence of PMA-activated neutrophils suggests gradual generation of the endogenous thrombin and prolonged but enhanced platelet activation.

Unlike the durations, the rates of each kinetic phase were less consistent and had a different dependency on the presence of activated neutrophils. Regression analyses conducted on the kinetic curves of contraction obtained in whole blood clots (Figure 4A) revealed that in the presence of PMA-activated neutrophils the average rate constants of phases 2 and 3 were reduced compared to non-activated neutrophils (Appendix A). The same trend was revealed in the kinetics of contraction in PRP clots (Appendix A), altogether indicating facilitated mechanical compaction of the clots in the presence of activated neutrophils. Since phases 2 and 3 are largely mechanical, i.e., reflecting progressive clot compression and mechanical stabilization [13], the observed differences in the phase rates suggest changes in the mechanical properties of fibrin in the presence of activated neutrophils.

### 3.3. PMA-Activated Neutrophils Produce NETs

Treating isolated neutrophils with 100 nM PMA at 37 °C for 3.5 h caused their activation, as assessed by the surface expression of active integrin Mac-1 (CD11b) measured by flow cytometry (Appendix A). Within the neutrophil gate of CD16+ signals taken as 100%, the average fraction of neutrophils expressing active CD11b in the control untreated samples was only 0.9 ± 1.3% (Appendix A), while in the PMA-treated samples, the average fraction of neutrophils expressing the active integrin Mac-1 reached 49 ± 16% (*p* = 0.03) (Appendix A) indicating strong activation of neutrophils by PMA.

Most importantly, under the conditions applied, the PMA-activated neutrophils formed NETs, which could be visualized directly either (immuno)histochemically after staining for histones and DNA (Appendix A) or with scanning electron microscopy (Appendix A). As a valid sign of NETosis in the PMA-activated neutrophil preparations, NET-specific citrullinated histones H3 and DNA were clearly detected as extracellular structures (Appendix A). The scanning electron micrographs captured an early stage of NETosis with a nascent NET (Appendix A) and the end of NETosis, where a free NET was no longer connected to a neutrophil (Appendix A). NETs could be visualized not only amid activated neutrophils, but also in the supernatant of a PMA-treated neutrophil preparation (Appendix A). Remarkably, the high-resolution scanning electron microscopy showed that the plasma membrane of PMA-treated neutrophils was highly porous (Appendix A), confirming that uncondensed chromatin, the structural basis of NETs, could be extruded through these pores, as shown earlier [16,17].

### 3.4. The Effects of Activated Neutrophils on Clot Contraction Are Associated with NETs Embedded in a Clot

To determine whether the observed effects of PMA-activated neutrophils on clot contraction are due to the NETs revealed in the neutrophil preparations, we first tried to see whether NETs exist in PRP clots formed in the presence of the activated neutrophils. Figure 5 provides direct evidence that NETs are incorporated into the structure of the clot, since they could be visualized histologically with and without fluorescence (Figure 5A–F) as well as with scanning electron microscopy (Figure 5G,H), although sometimes it is hard to clearly distinguish the fibrillar structures of NETs from fibrin [18]. These findings show that blood clotting in the presence of neutrophils pre-activated for a few hours with PMA results in the embedding of NETs into the clot, strongly suggesting that the observed effects of PMA-activated neutrophils on clot contraction are associated with NETs.

The hypothesis of the key role of NETs in clot contraction has been confirmed by experiments in which NETs in a clot were destroyed by enzymatic cleavage with deoxyribonuclease (DNAse) I (incubation for 30 min at 37 °C). The effects of DNAse I on clot contraction in the absence or presence of PMA-activated neutrophils are shown for whole blood (Figure 6, Appendix A) and PRP (Appendix A, Appendix A). DNAse I did not induce any changes by itself, but it completely abolished the stimulating effect of PMA-activated neutrophils on the final extent and average rates of clot contraction in whole blood (Figure 6A,D; Appendix A). A similar inhibitory effect of DNAse I on clot contraction was observed in PRP-clots in which contraction was initially enhanced with activated neutrophils but overturned by DNase I, as assessed by the final extent of clot contraction and area under the kinetic curve (Appendix A; Appendix A). The absence of the effect of DNAse I on the average rate in PRP, unlike in whole blood, can be explained by the fact that the rate of contraction in PRP was >2 times faster, which makes the relative inhibitory effect harder to reveal. Taken together, these findings indicate that the stimulating effects of PMA-activated neutrophils on clot contraction are associated with NETs incorporated into the whole blood or plasma clots.

### 3.5. The Promotion of Clot Contraction by NETs Is Mediated by Enhanced Generation of Endogenous Thrombin

Based on the stimulating effect of PMA-activated neutrophils on the initiation of clot contraction during phase 1 (Figure 4), a conceivable mechanism underlying the amplification of blood clot contraction by NETs is the enhancement of endogenous thrombin production, causing amplified secondary platelet activation and enhanced contractility. To test this assumption, we used rivaroxaban, an inhibitor of factor Xa capable of preventing the formation of endogenous thrombin. Rivaroxaban was added to whole blood (Figure 7, Appendix A) or PRP (Appendix A, Appendix A) containing non-activated or PMA-activated neutrophils before thrombin-induced clot formation and contraction. Unlike with non-activated neutrophils, in the presence of rivaroxaban the final extent of clot contraction, area under the kinetic curve, and average rate were reduced significantly in whole blood with the PMA-activated neutrophils (Figure 7, Appendix A). Similar effects were observed in PRP samples (Appendix A, Appendix A), such that in the presence of rivaroxaban the final extent of clot contraction and average rate were substantially lowered with PMA-activated neutrophils, while there were no visible effects of rivaroxaban in the presence of non-activated neutrophils (Appendix A; Appendix A).

Therefore, suppression of endogenous thrombin generation via inhibition of factor Xa abrogates the stimulating effect of NETs on clot contraction. The results obtained suggest that in the presence of PMA-activated neutrophils that form NETs, the enhanced clot contraction is due to increased production of endogenous thrombin that hyperactivates platelets, making them more contractile [19].

### 3.6. NETs Make the Fibrin Clot Softer

As suggested by the variations in the kinetics of the mechanical stages of clot contraction during phases 2 and 3 (Figure 4, Appendix A), another conceivable mechanism underlying the promoting effect of PMA-activated neutrophils on clot contraction is the effect of NETs on the clot mechanical properties or clot deformability. To test this assumption, we utilized a common technique and apparatus to study clot elasticity, namely thromboelastography (TEG), which records changes in clot stiffness over time in response to shear. In addition to the blood clotting time (*R* in seconds), TEG provides information about a clot mechanical feature known as the maximal clot firmness (*MA*, maximal amplitude in millimeters), which can be converted to absolute clot stiffness or elasticity (storage modulus, *G’*, in Pascal). First, we looked at the effects of PMA-activated and non-activated neutrophils on the clotting time and clot stiffness in whole blood and PRP. In whole blood clots, there was no noticeable influence of PMA-activated neutrophils on the parameters of TEG (Figure 8A,B; Appendix A), while in PRP clots, NETs slightly prolonged the clotting time and, most importantly, reduced the final stiffness of the clot (Figure 8C,D; Appendix A), indicating that NETs are able to make the clot softer.

The link between clot stiffness and NETs was additionally tested by applying DNAse I and assessing the effects of NET cleavage on TEG parameters in whole blood and PRP in the presence of PMA-activated neutrophils. Before thrombin-induced initiation of clotting and registration of TEG, blood or PRP samples were mixed with untreated or PMA-activated neutrophils, followed by addition of DNAse I and incubation for 30 min at 37 °C (Figure 9, Appendix A). Treatment with DNAse I increased the average storage modulus (G’) in whole blood clots with PMA-activated neutrophils, suggesting that NETs were responsible for the reduced stiffness of the clot. Accordingly, there was no effect of DNAse I in the blood samples containing non-activated neutrophils (Figure 9B, Appendix A). Surprisingly, unlike in whole blood, in PRP, addition of DNAse I did not affect the TEG parameters (Figure 9C,D; Appendix A), perhaps due to the much stiffer plasma clots with a higher volume fraction of fibrin compared to whole blood, as seen from the maximal values on the Y-axes in Figure 9B,D.

## 4. Discussion

Inflammatory thrombosis or immunothrombosis is a life-threatening complication in which dysregulated co-activation of the immune and blood clotting systems leads to the formation of obstructive venous or arterial thrombi with a strong inflammatory component [20,21]. This dysregulation is a pathogenic mechanism developing in a wide range of severe inflammatory conditions [22,23]. The clinical course and outcomes of inflammatory thrombosis, including severe organ damage and thromboembolic events, as well as efficacy of treatments, are largely dictated by the composition and dynamic properties of the resultant thrombus, such as mechanical stability, permeability, and susceptibility to lysis [24]. One of the major determinants of these properties is the maturation and remodeling of the thrombus, known as platelet-driven clot contraction (also known as retraction) [12]. Although the fundamental role of platelets, RBCs and plasma proteins in modulating clot contraction is well-established [13], the impact of leukocytes, especially activated neutrophils embedded in the blood clot, remains unclear. To address this question directly, we have employed a model system of neutrophil-enriched human blood or plasma clots, enabling us to analyze how neutrophil activation and NETosis affect the clot contraction process. Importantly, for whole blood, the already present neutrophils were non-PMA-activated and therefore they had no effect on clot contraction via NET production. A potential activation of endogenous neutrophils associated with the contamination of exogenous neutrophil preparations with PMA has been excluded (Figure 3, Appendix A). A potential background effect of the already present neutrophils was also excluded by the negative controls without addition of neutrophils and with addition of exogenous non-activated neutrophils (Figure 1, Figure 2, Figure 6 and Figure 8, and Appendix A).

The main finding of this study is that activated neutrophils promote the contraction of blood clots, and this effect is directly related to the formation of NETs embedded within the fibrin network. Clot contraction is enhanced substantially in the presence of PMA-stimulated neutrophils (Figure 1) producing extracellular traps, which comprise uncondensed chromatin built of DNA and histones (Appendix A and Figure 2). We have provided direct evidence that NETs are incorporated into the structure of a fibrin clot, since they have been visualized using histochemical techniques (Figure 5A–F), as well as with scanning electron microscopy (Figure 5G,H). Notably, the stimulating effect of PMA-activated neutrophils on clot contraction in both whole blood and PRP (Figure 6A,D and Appendix A) has been shown to be abolished by treatment with DNase I, which catalyzes cleavage and elimination of NETs. Taken together, these results can indicate that NETs are essential for the promoting effect of the activated neutrophils on clot contraction. Accordingly, while the synergistic contributions from other secretory products released by activated neutrophils (serine proteases, reactive oxygen species (ROS), etc.) cannot be ruled out completely, the rescue of “normal” clot contraction with DNase I strongly suggests that NET formation is the dominant mechanism. Hypothetically, neutrophil elastase can cleave fibrinogen and potentially alter clot architecture [25], while certain ROS can activate platelets [26]. Future work using specific inhibitors of degranulation or oxidative burst could help delineate any potential synergistic or antagonistic effects of these pathways relative to the dominant pro-contractile signal provided by NETs. Our new data are in line with the growing evidence identifying NETs not merely as ubiquitous components of thrombi but as active, strong (pro)thrombotic structures that promote formation of stable thrombi in various conditions, including deep vein thrombosis, acute ischemic stroke, and myocardial infarction [4,6,22,27,28,29,30].

Trying to decipher the underlying mechanisms, we have identified at least two distinct pathways through which NETs facilitate clot contraction: (i) by enhancing the local production of endogenous thrombin and (ii) by affecting the mechanical properties of the fibrin network within inflammatory clots.

It has been shown that the histone–DNA complexes within NETs potentiate thrombin generation via platelet-dependent and platelet-independent mechanisms [31,32]. Since thrombin is a potent platelet activator known to enhance their contractility [8,13,33], we assumed that prevention of the NET-induced local generation of thrombin within the clot would reduce or abrogate the promoting effect of the activated neutrophils on clot contraction. Indeed, in the presence of rivaroxaban (an inhibitor of factor Xa), the rate and extent of clot contraction in the presence of PMA-activated neutrophils have been significantly reduced to the level of control non-activated neutrophils (Figure 7A,C,D and Appendix A), strongly suggesting the critical role of NET-mediated endogenous thrombin generation as a mechanism promoting contraction of inflammatory thrombi. This observation is in line with the findings in which activated monocytes promote clot contraction via expression of tissue factor and local thrombin generation, the most powerful physiological platelet agonist promoting platelet contractility [33]. This observation is also in line with publications showing that NETs promote blood clotting [4,34,35,36].

As an alternative or additional mechanism for boosting clot contraction by the activated neutrophils, we have found that NETs have altered the mechanical properties of clots, decreasing their stiffness, as determined by rheological measurements using TEG (Figure 8D and Figure 9B). Accordingly, the enzymatic cleavage of NETs with DNase I has restored the clot firmness, although this effect was more pronounced in whole blood compared to PRP clots (Figure 9C,D). Altogether, these data suggest that NETs can soften blood clots, making the fibrin scaffold less resilient to the platelet-generated contractile forces, thus facilitating clot shrinkage and compaction. A similar conclusion was drawn from the effects of DNA and histones on the mechanical stability of fibrin clots formed by staphylocoagulase [37]. It is noteworthy that the observed softening of plasma clots in the presence of NETs seemingly contradicts the results obtained by Longstaff et al. [38], showing that fibrin clots formed in the presence of the mixture of histones and DNA were more resistant to shear forces. However, when added separately, DNA and histones had opposing effects on fibrin stiffness, such that in the presence of DNA alone, fibrin was more sensitive to mechanical shear. This contradiction suggests complex relations between the components of NETs and fibrin mechanics depending on the concentrations, applied shear stresses, and perhaps other conditions. It is conceivable that clot formation kinetics affected by NETs contribute to the diversity of clot structure which is a major determinant of blood clot mechanics [37,38].

The enhanced clot contractility under the influence of activated neutrophils and NETs has important pathophysiological implications, indirectly affecting at least three key pathogenic features of in vivo thrombi: obstructiveness, susceptibility to fibrinolysis, and embologenicity. Firstly, the increased contraction of NET-enriched thrombi is likely to reduce the overall clot volume, thereby decreasing the degree of vascular occlusion. This might be relevant to inflammatory venous thrombosis (e.g., sepsis-associated or cancer-associated deep vein thrombosis), where highly compacted thrombi may cause less severe flow obstruction compared to voluminous thrombi [39]. Secondly, densification of fibrin during clot contraction is known to increase clot’s susceptibility to endogenous fibrinolysis due to the proximity of fibrin fibers to each other and an increase in the t-PA-to-fibrin ratio [40]. Presumably, NET-enhanced contraction of inflammatory thrombi can accelerate natural fibrinolysis, promoting faster resolution and reducing the thrombus durability [41,42,43]. The latter idea is supported by the data showing that following the initial lytic stability, NET-rich fibrin network is ultimately cleaved by plasmin, suggesting self-restraint of inflammatory thrombosis [38,44]. Thirdly, the embologenicity of inflammatory thrombi is potentially diminished because a well-contracted blood clot is more stable mechanically and less prone to rupture and fragmentation [45,46]. In summary, although inflammation creates a hypercoagulable state that promotes thrombosis, the resulting NET-enhanced contraction may reduce the risk of thrombotic embolization and support gradual dissolution of a thrombus rather than its accidental breakdown [4,31].

The severity of inflammation, and, consequently, the degree of activated neutrophils and NETs involvement, can be of great clinical importance for treatment efficacy. Our data suggest that inflammatory NET-rich thrombi are likely prone to enhanced contraction, resulting in denser occlusive thrombi that may be less susceptible to thrombolytic therapy and mechanical thrombectomy. The reinforced compaction and low porosity of NET-rich thrombi may reduce the efficacy of therapeutic thrombolysis, since contracted clots are less sensitive to the external fibrinolysis [38,40,47,48,49]. It has been shown that embedding histone–DNA complexes into fibrin results in the higher resistance of fibrin clots to external lysis [37,38]. Since the enhanced clot contraction results in formation of mechanically strong and stiff thrombi [27,50,51], it may be more difficult to remove an inflammatory NET-containing thrombus during mechanical thrombectomy.

The results of this study should be considered within the context of several limitations, namely: (1) the study was conducted on the in vitro model, which limits its pathophysiological implications; (2) a single stimulant, PMA, was used to activate neutrophils, which reproduces partially physiological reactions; (3) the evidence for endogenous thrombin generation induced by NETs was indirect (with the use of inhibitor rivaroxaban) because of the presence of exogenous thrombin that would affect direct measurements of thrombin activity; (4) a relatively small sample size, which, in combination with the variability of individual blood samples, could have led to a moderate statistical significance, albeit without affecting the reliability of the main results and conclusions.

Based on the peculiarities of mechanical and structural rearrangement of inflammatory thrombi, assessing the level of soluble NET biomarkers in the blood [22,30,52] and/or developing imaging techniques to detect NET-rich thrombi [53,54] could improve diagnostics and help with the risk prognosis in immunothrombosis [55,56]. Potentially, adjuvant therapies aimed at cleaving NETs with DNase I or suppressing NETosis with inhibitors of peptidylarginine deiminase type 4 (PAD4), an enzyme that catalyzes citrullination of H3 histones essential for NETosis [57,58], may improve the outcomes of immunothrombosis by rendering inflammatory thrombi more susceptible to therapeutic thrombolysis and mechanical thrombectomy [59]. Targeting NETosis with PAD4 inhibitors offers promising results in inflammatory and thrombotic diseases, but carries a significant risk of off-target immune suppression due to disruption of dendritic cell and T-cell function. This suggests that while NETosis inhibition may reduce acute inflammation, it may impair the adaptive immunity necessary for pathogen clearance, making the rationale for clinical application challenging [60].

## 5. Conclusions

Activated neutrophils promote blood clot contraction via the formation of NETs. Clinically, the enhancement of clot contraction under the influence of inflammatory cells may have both positive and negative aspects. The positive consequences include those related to the pathophysiology of immunothrombosis, namely, reduced thrombus obstructiveness, increased susceptibility to intrinsic lysis, and decreased embologenic potential. Conversely, the negative aspects of enhanced contractility in an inflammatory thrombus include difficulties in its removal, specifically, reduced susceptibility to exogenous thrombolysis and more challenging mechanical thrombectomy.

## Figures and Tables

**Figure 1 cells-14-02018-f001:**
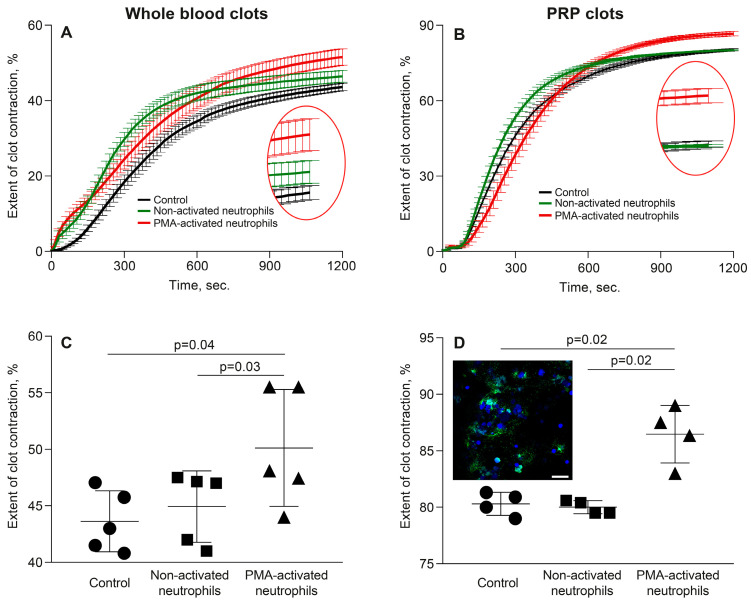
Phorbol-12-myristate-13-acetate (PMA)-activated neutrophils, unlike non-activated neutrophils, promote clot contraction. Thrombin-induced clots were formed in whole blood (**A**,**C**) or platelet-rich plasma (PRP) (**B**,**D**) in the absence (control) and presence of non-activated or PMA-activated neutrophils followed by optical tracking of the clot size. The results are presented as the averaged kinetic curves (**A**,**B**) and the final extent (**C**,**D**) of clot contraction. (**D**) Inset: Fluorescent microscopy of PMA-activated neutrophils within a PRP clot stained for citrullinated histones CitH3 (*green*) and deoxyribonucleic acid (DNA) (*blue*). Magnification bar 20 µm. Other kinetic parameters of clot contraction are shown in Figure 2. Data points in the curves and the values of the extent of clot contraction represent the mean ± SD (n = 5 for blood, n = 4 for PRP, individual donors). One-way ANOVA was used for comparisons. Only significant differences are shown.

**Figure 2 cells-14-02018-f002:**
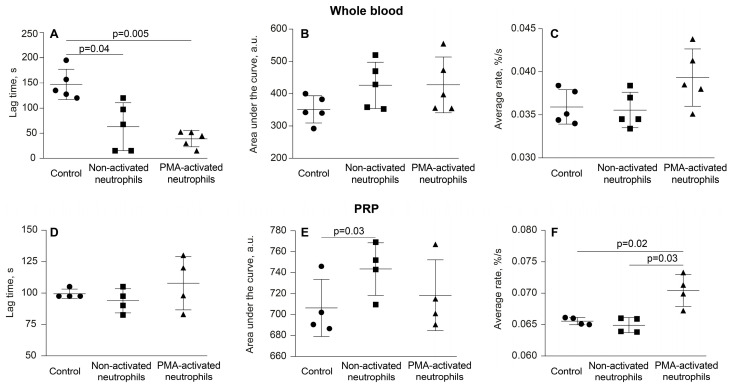
PMA-activated neutrophils, unlike non-activated neutrophils, enhance the kinetic parameters of clot contraction in whole blood and PRP. Clot formation and contraction were induced by 1 U/mL thrombin followed by the optical tracking of the clot size. The lag time (**A**,**D**), area under the curve (**B**,**E**) and average rate (**C**,**F**) were measured in clots formed in whole blood (n = 5) and PRP (n = 4) samples obtained from independent donors. Results are presented as the mean ± SD. One-way ANOVA was used for comparisons. Only significant differences are shown. For numerical data and detailed statistical analysis see Appendix A.

**Figure 3 cells-14-02018-f003:**
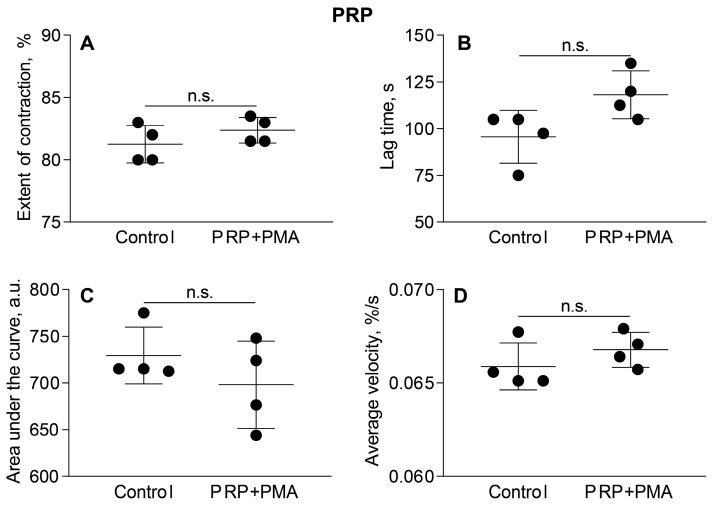
PMA in the residual concentration added with PMA-activated neutrophils (22 nM PMA) does not affect by itself the parameters of clot contraction in PRP. Clot formation and contraction were induced by 1 U/mL thrombin followed by the optical tracking of the clot size. The final extent of clot contraction (**A**), lag time (**B**), area under the curve (**C**) and average rate (**D**) were measured in clots formed in PRP samples obtained from independent donors (n = 4). Results are presented as the mean ± SD. Student’s paired *t*-test was used for comparisons. Abbreviation: n.s.—not significant. For numerical data and detailed statistical analysis see Appendix A.

**Figure 4 cells-14-02018-f004:**
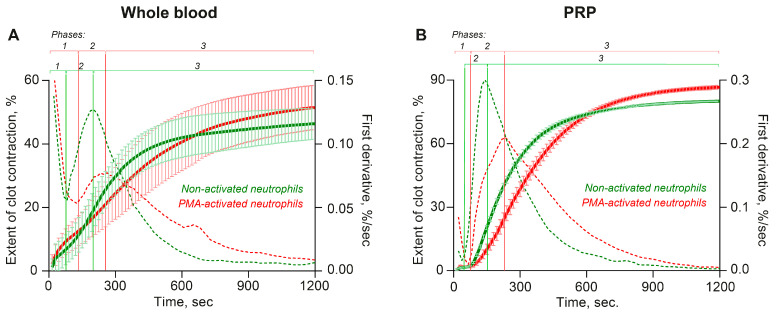
PMA-activated neutrophils, unlike non-activated neutrophils, affect parameters of the kinetic phase analysis of the averaged kinetic curves of contraction of clots formed in whole blood (**A**) and PRP (**B**). Clot formation and contraction were induced by 1 U/mL thrombin followed by the optical tracking of the clot size. On the averaged kinetic curves, transitions between the phases of contraction (vertical solid lines) were determined by finding local maxima and minima within the instantaneous first derivatives (dashed curves). The kinetic curves were fit using a piecewise function, and the duration and rate constant of each phase were determined. The results are presented as the mean ± SD (n = 5 for whole blood and n = 4 for PRP using the blood of independent donors). For numerical data and detailed statistical analysis see Appendix A.

**Figure 5 cells-14-02018-f005:**
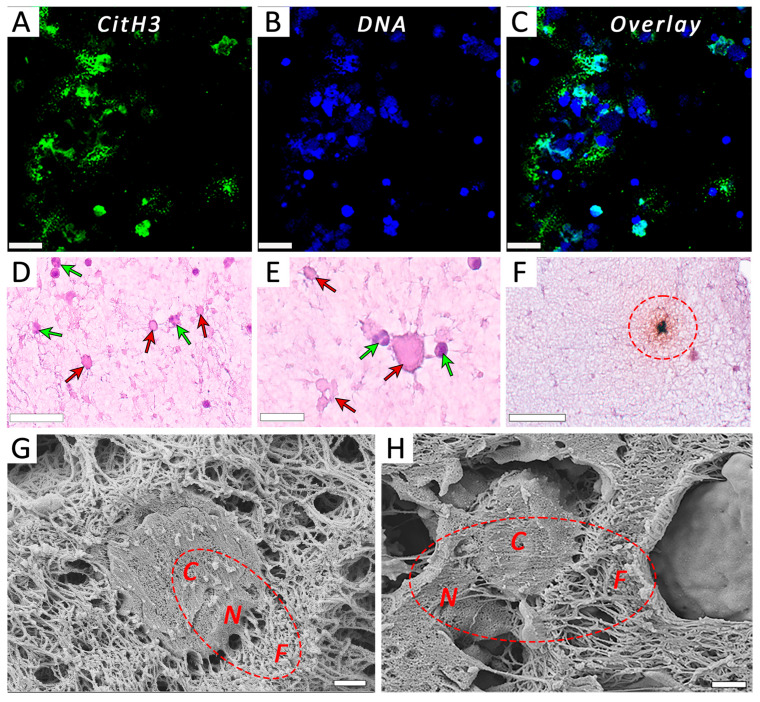
Neutrophil extracellular traps (NETs) released from PMA-activated neutrophils are embedded within PRP clots as revealed by light microscopy and scanning electron microscopy. (**A**–**C**) Fluorescent microscopy of PMA-activated neutrophils within a PRP clot stained for citrullinated histones CitH3 (**A**), DNA (**B**), and overlay (**C**). (**D**,**E**) Non-activated, mainly segmented nuclear neutrophils (*green arrows*) and PMA-activated predominantly enucleated neutrophils of irregular shapes with indistinct cell membranes and/or pericellular structures (*red arrows*) within a PRP clot. (**F**) A PMA-activated neutrophil within a clot stained for citrullinated histones CitH3 (*dashed circle*) on top of hematoxylin & eosin stain to visualize the cell nucleus and pericellular uncondensed chromatin (NETs). (**G**,**H**) Representative scanning electron micrographs of PRP clots showing the incorporation of PMA-activated neutrophils and surrounding fibrillar structures that comprise NETs and fibrin. Unlike fibrin, NETs are characterized by a higher network density and lower porosity as well as thinner filaments and typical co-localization with neutrophils. Designations: *C*—cell, *N*—NETs, *F*—fibrin. Magnification bars: 20 µm (**A**–**C**), 50 µm (**D**,**F**), 25 µm (**E**), and 1 µm (**G**,**H**).

**Figure 6 cells-14-02018-f006:**
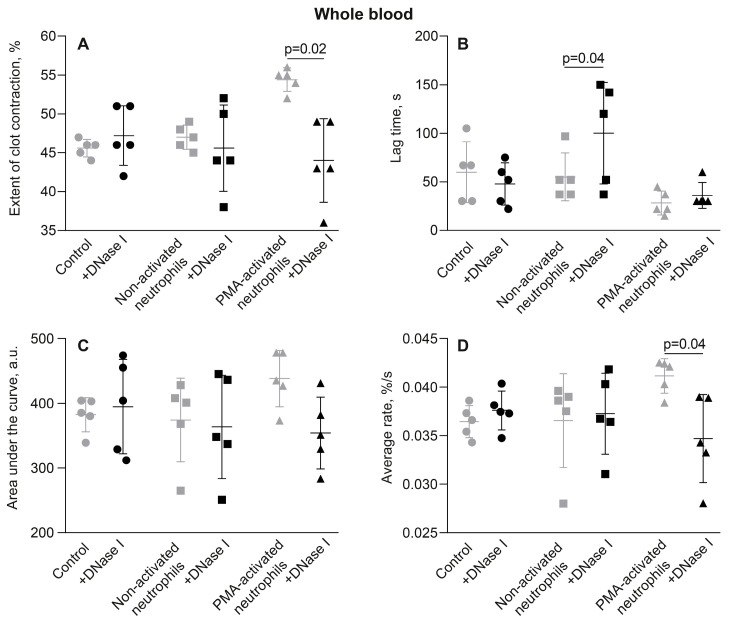
Deoxyribonuclease (DNase) I abrogates the promoting effects of PMA-activated neutrophils on clot contraction, but has no effect in the presence of non-activated neutrophils or by itself. The effects of DNase I on the parameters of clot contraction were studied in whole blood clotted in the absence (control) and presence of non-activated or PMA-activated neutrophils. Clot formation and contraction were induced by 1 U/mL thrombin followed by the optical tracking of a clot size. The final extent of clot contraction (**A**), lag time (**B**), area under the curve (**C**), and average rate (**D**) were measured in clots formed in blood samples obtained from independent donors. Results are presented as the mean ± SD (n = 5). Student’s paired *t*-test was used for comparisons. Only significant differences are shown. For numerical data and detailed statistical analysis see Appendix A.

**Figure 7 cells-14-02018-f007:**
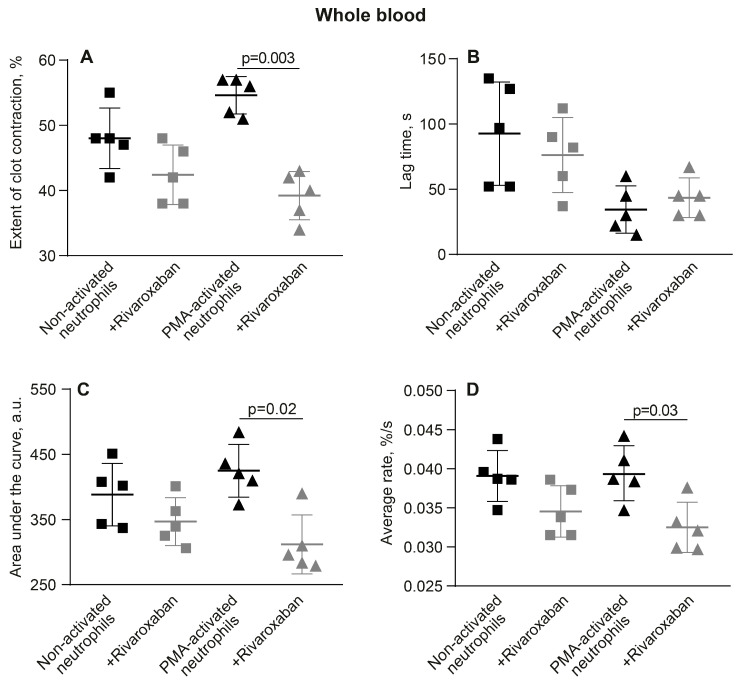
Rivaroxaban abrogates the promoting effects of PMA-activated neutrophils on blood clot contraction, but has no effect in the presence of non-activated neutrophils. Clot formation and contraction were induced by 1 U/mL thrombin followed by the optical tracking of the clot size. The final extent of clot contraction (**A**), lag time (**B**), area under the curve (**C**), and average rate (**D**) were measured in clots formed in blood samples obtained from independent donors. Results are presented as the mean ± SD (n = 5). Student’s paired *t*-test was used for comparisons. Only significant differences are shown. For numerical data and detailed statistical analysis see Appendix A.

**Figure 8 cells-14-02018-f008:**
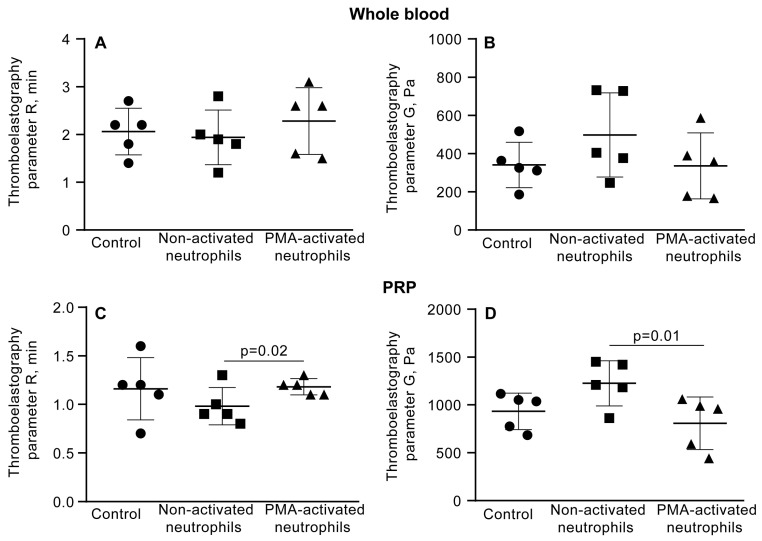
PMA-activated neutrophils, unlike non-activated neutrophils, reduce the shear elastic modulus (*G’*) of PRP clots. Clot formation and contraction were induced by 1 U/mL thrombin in whole blood (**A**,**B**) or PRP (**C**,**D**) obtained from independent donors. Results are presented as the mean ± SD (n = 5). One-way ANOVA. Only significant differences are shown. For numerical data and detailed statistical analysis see Appendix A.

**Figure 9 cells-14-02018-f009:**
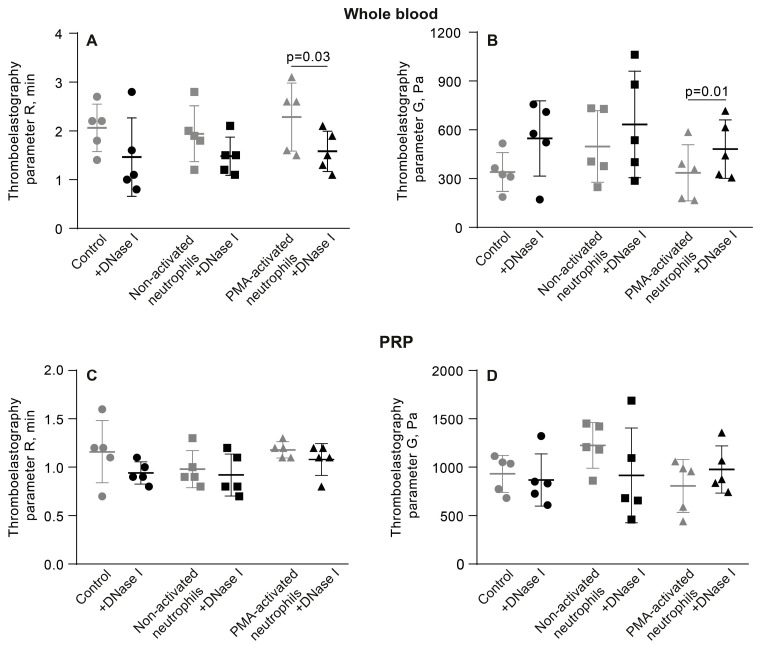
DNase I abrogates the softening effect of PMA-activated neutrophils on PRP clots, but has no effect in the presence of non-activated neutrophils or by itself. Effects of DNase I on the parameters of TEG, the reaction time (*R*) and clot elastic modulus (*G’*), in the absence (control) and presence of non-activated or PMA-activated neutrophils. Clot formation and contraction were induced by 1 U/mL thrombin in whole blood (**A**,**B**) or PRP (**C**,**D**) obtained from independent donors. Results are presented as the mean ± SD (n = 5). Student’s paired *t*-test was used for comparisons. Only significant differences are shown. For numerical data and detailed statistical analysis see Appendix A.

## Data Availability

The original contributions presented in this study are included in the article/Appendix A. Further inquiries can be directed to the corresponding author.

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
