# Peer review of "Neutrophil Extracellular Traps Promote Platelet-Driven Contraction of Inflammatory Blood Clots via Local Generation of Endogenous Thrombin and Softening of the Fibrin Network"

_cells, 2025, doi:10.3390/cells14242018_

Round 1
Reviewer 1 Report
Comments and Suggestions for Authors
This paper presents the effects of NETs on blood contraction. This is an original idea, and of interest, since NETs are now regularly identified for a role in the coagulation system. There are some questions when I was reading the paper.
L.97: You add neutrophils to the whole blood (0.8*106 to 500 µL), which gives a concentration that is quite similar to the concentration in already present in whole blood (1.5-9*106/ml). The already present neutrophils may have affected your results and should be discussed.
L.107: You collect the supernatant of PMA-activated neutrophils after gravitational sedimentation of the cells. Since the NETs form a network, it needs to be confirmed what the level of the NETs is in the supernatant, since I could imagine that large networks would precipitate with the neutrophils. Also when the NETs are still attached to the neutrophils, they will end up in the sediment. There are commercial assays available to determine the levels of NETs (MPO-DNA or cH3) and quantification of the NETs in the supernatant and suspensions would be relevant information. Another question is about the goal to study both a suspension and a supernatant. Can you describe this in more detail. For example, do you expect differences in NETs that are connected or detached from neutrophils?
L 114: you use CD11b as a marker for neutrophil activation. Is it a specific marker for PMA-induced activation?
L.129: Please, provide information on how was the platelet count was performed?
L 127): NETs form a very fragile network, how did you ensure and check that it is still intact in the clots used for contraction?
L.171: Why were NETs observed only in PFP clots and not in PRP clots, which are more comparable to the other experimental conditions?
L.191: Minor correction in the text: “8×105” should be replaced with “10⁵.”
L 254: PMA is described to have other effects on neutrophils than induction of NETosis. It would be informative to discuss this. Furthermore, you sometimes talk about activated neutrophils and NETs, but the presence and concentration of NETs is not always clear. Please, be consistent on the use, and make clear why you use which word, is it because you think there may be other effects than by NETs?
L267: you mention here that the PMA concentration used is 2.2 mM, while in the methods you describe 100 mM. To me it is not clear where this difference comes from.
Figure 2:
- Why were non-activated neutrophils not included in the SEM imaging?
- When I look at A-C compared to D-F, I would expect more cells and NETs in the latter series, compared to the scale of A-C.
- How were fibrin fibers distinguished from NETs in the SEM images? I am not convinced you can distinguish them clearly. The SEM pictures are however, very beautiful.
Figure 3: It would be of interest to also compare activated and non-activated neutrophils, and whether the activated neutrophils are similar to non-activated neutrophils with DNAse. Also in this figure, as well as in the other figures, sometimes there is little difference between the significant and non-significant findings/trends if you look at the effects. With so few measurements, it would be good to be more careful about the conclusions.
L 286: you propose that endogenous thrombin is generated. One way to check that is by using a chromogenic thrombin substrate. Did you test this, or use another method? Since NETs have a negatively charged surface, I could imagine activation of the contact activation, leading to thrombin generation. It would be nice to discuss and study this.
L.350: Additional details should be provided regarding the timing of DNase I addition to the samples.
Figure 3: How can the absence of an effect of DNase I on the average rate in PRP be explained, while an effect is seen in whole blood?
Figure 4: In whole blood, how do the authors explain the apparent lack of difference between activated and non-activated neutrophils in the extent of clot contraction?
It would also be helpful to add “Whole Blood” and “PRP” labels directly to Figures 4 and 6, since they are referenced in Figure 5.
Why is there no control condition included in this figure?
TEG results: The effects of DNase on clot stiffness occur on different parameters than those affected by PMA-activated neutrophils. Therefore, the authors’ conclusion that DNase reverses the effects of NETs on clot stiffness appears premature and would require additional experimental evidence.
Reviewer 2 Report
Comments and Suggestions for Authors
This manuscript investigates how neutrophils and NETs influence platelet-driven clot contraction by combining microscopy and mechanical assessment of thrombin-induced clots. The authors report that activated neutrophils increase the rate and extent of contraction through NET-dependent mechanisms and propose that these findings may have therapeutic implications. Although the study addresses an important aspect of immunothrombosis, several issues need clarification, as outlined below.
- The results subsection titles read more like methods descriptors and should be revised so that each one communicates the main finding of the section.
- Representative fluorescence microscopy images would be more effectively presented within the quantitative figure panels so that visual and numerical evidence are directly aligned.
- The addition of a biochemical NET readout such as MPO–DNA complexes would strengthen the validation of NETosis and provide an orthogonal confirmation beyond microscopy.
- Statistical annotations appear in inconsistent positions across figures, with some placed above and others below the plotted values. A consistent layout would improve readability.
- Figures would be less visually crowded if the authors limited significance markers to comparisons that are statistically significant, rather than displaying ns indicators.
- The exclusive use of PMA to induce NETosis raises questions about generalisability. The authors should discuss this limitation or consider validating key findings with a more physiological NETosis stimulus.
- The discussion of therapeutic relevance is brief and would benefit from clearer justification of its clinical translation. This should include a more detailed consideration of NETosis targeting with specific attention to PAD4 inhibition and the risk of off-target immune effects (DOI 10.1016/j.mucimm.2025.04.006).
Reviewer 3 Report
Comments and Suggestions for Authors
Dear Authors,
General comment
The manuscript presents a well-designed and timely study addressing the impact of neutrophil extracellular traps (NETs) on platelet-driven clot contraction, an area of increasing interest within the context of immunothrombosis. The combination of kinetic analysis, imaging, and thromboelastography provides valuable insights into how activated neutrophils modulate clot dynamics. The findings suggesting that NETs enhance clot contraction through endogenous thrombin generation and altered clot mechanics are scientifically relevant and may contribute to a better understanding of inflammatory thrombus behavior. However, several methodological limitations and interpretational aspects should be clarified to strengthen the manuscript and improve its suitability for publication in Cells.
Major comments
-
The use of PMA as the sole stimulus for NET formation raises concerns regarding physiological relevance. PMA induces a strong and artificial NETosis pathway that does not fully reflect NET formation during infection or sterile inflammation. The authors should include a more detailed justification for this model and discuss potential differences compared to NETs induced by more physiologically relevant stimuli (e.g., LPS, IL-8, calcium ionophore).
-
The mechanistic link between NETs and enhanced endogenous thrombin generation is central to the proposed model; however, no direct measurement of thrombin generation is provided. Inclusion of thrombin generation assays or surrogate markers (e.g., F1+2, TAT complexes) would significantly strengthen the conclusions. At minimum, this limitation should be acknowledged in the Discussion.
-
The interpretation of TEG data suggesting that NETs soften clots requires caution. TEG provides limited mechanical characterization, and the conclusion that NETs decrease clot stiffness may be premature. The authors should either provide additional mechanical analyses (e.g., rheometry, AFM) or moderate the interpretation.
-
DNase I experiments are interpreted exclusively as NET degradation effects. DNase I has been reported to influence fibrin architecture independently of NETs, potentially altering clot mechanics. This confounding factor should be addressed, and the authors should clarify whether DNase I treatment may have affected fibrin organization.
-
The manuscript repeatedly extrapolates the in vitro findings to in vivo thrombosis and clinical implications. Given the absence of in vivo validation, statements regarding thrombus obstructiveness, embolization risk, and therapeutic implications should be tempered.
Minor comments
-
The introduction is lengthy and could be streamlined by focusing more directly on the knowledge gap regarding clot contraction and NET involvement.
-
The description of donor characteristics lacks information on potential confounders such as smoking status or recent infections, which may influence neutrophil function.
-
Figure 2 would benefit from improved labeling and contrast to distinguish NET structures from fibrin fibers.
Round 2
Reviewer 1 Report
Comments and Suggestions for Authors
Most comments are in the revised manuscript, but some comments remain: see added word file (cells-4024150-review_v2.docx).

Reviewer 2 Report
Comments and Suggestions for Authors
The authors have successfully addressed all my concerns.
Author Response
We are pleased that Reviewer #2 is satisfied with our responses.
Reviewer 3 Report
Comments and Suggestions for Authors
Dear Authors,
You have succesfully addresed the majority of my comments. Well done.
Author Response
We are pleased that Reviewer #3 is satisfied with our responses.
Round 3
Reviewer 1 Report
Comments and Suggestions for Authors
My comments are all incorporated in the last version, I have no other questions.